# Active Symbolic Discovery of Ordinary Differential Equations via Phase Portrait Sketching

**Nan Jiang[1], Md Nasim[2], Yexiang Xue[1]**

[1]Department of Computer Science, Purdue University, USA
[2]Department of Computer Science, Cornell University, USA
{jiang631, yexiang}@purdue.edu, md.nasim@cornell.edu

## Abstract

The symbolic discovery of Ordinary Differential Equations (ODEs) from trajectory data plays a pivotal role in AI-driven scientific discovery. Existing symbolic methods predominantly rely on fixed, pre-collected training datasets, which often result in suboptimal performance, as demonstrated in our case study in Figure 1. Drawing inspiration from active learning, we investigate strategies to query informative trajectory data that can enhance the evaluation of predicted ODEs. However, the butterfly effect in dynamical systems reveals that small variations in initial conditions can lead to drastically different trajectories, necessitating the storage of vast quantities of trajectory data using conventional active learning. To address this, we introduce **A**ctive Symbolic Discovery of Ordinary Differential Equations via **P**hase **P**ortrait **S**ketching (APPS). Instead of directly selecting individual initial conditions, our APPS first identifies an informative region within the phase space and then samples a batch of initial conditions from this region. Compared to traditional active learning methods, APPS mitigates the gap of maintaining a large amount of data. Extensive experiments demonstrate that APPS consistently discovers more accurate ODE expressions than baseline methods using passively collected datasets.

**Code** — https://github.com/jiangnanhugo/APPS-ODE
**Extended version** — https://arxiv.org/abs/2409.01416

## 1 Introduction

Uncovering the governing principles of physical systems from experimental data is a crucial task in AI-driven scientific discovery (Schmidt and Lipson 2009; Zhang and Lin 2018; Wu and Tegmark 2019). Recent advancements have introduced various methods for uncovering knowledge of dynamical systems in symbolic Ordinary Differential Equation (ODE) form, leveraging techniques such as genetic programming (He et al. 2022), sparse regression (Brunton, Proctor, and Kutz 2016; Fasel et al. 2022), Monte Carlo tree search (Sun et al. 2023), pretrained Transformers (Qian, Kacprzyk, and van der Schaar 2022), and deep reinforcement learning (Jiang, Nasim, and Xue 2024).

State-of-the-art approaches discover the symbolic ODEs using a fixed, pre-collected training dataset. However, their

performance is often heavily influenced by the quality of the collected data. As illustrated in Figure 1, we find that the best-discovered ODEs from the most recent baseline, that is ODEFormer (d'Ascoli et al. 2024), may fit some test trajectories well, but fit other test trajectories poorly. This observation highlights the need for new methods that actively query informative trajectory data to improve ODE discovery.

Suppose trajectory data can be obtained from a data oracle by specifying the initial conditions. To minimize excessively querying the oracle, a key challenge emerges: given a set of candidate ODEs predicted by a learning method, how can initial conditions within the variable intervals be strategically selected to obtain informative data?

Previous work in the active learning literature typically maintains a large set of data, evaluates their informativeness, and then queries the most informative data points (Golovin, Krause, and Ray 2010; Medina and White 2023). However, the chaotic nature of dynamical systems complicates the direct application of such methods. The Butterfly effect states that small variations in initial conditions can lead to vastly different outcomes. For instance, as illustrated in Figure 2(c), selecting initial conditions near $(3, 0)$ for $\phi_1$ can result in trajectories that diverge in opposite directions. Effectively addressing this variability requires densely sampling initial conditions to thoroughly explore the space. Existing active learning-based approaches will be computationally prohibitive and demand significant memory resources, particularly in high-dimensional dynamical systems.

To address these challenges, we propose a novel approach to data querying. We consider selecting a batch of close-neighbor initial conditions instead of individual initial conditions. This process begins by sketching the dynamics in smaller regions, identifying an *informative region* in the phase space, and then sampling a batch of initial conditions from this region. Figure 2(c) illustrates this idea using phase portraits for three candidate ODEs. Region $u_2$ is chosen because the trajectories generated by the candidate ODEs exhibit greater divergence in this region than region $u_1$. Section 3 provides detailed region selection criteria.

Thus, we introduce **A**ctive Symbolic Discovery of Ordinary Differential Equations via **P**hase **P**ortrait **S**ketching (APPS), which consists of two key components: (1) a deep sequential decoder, which guides the search for candidate ODEs by sampling from the defined grammar rules. (2) a

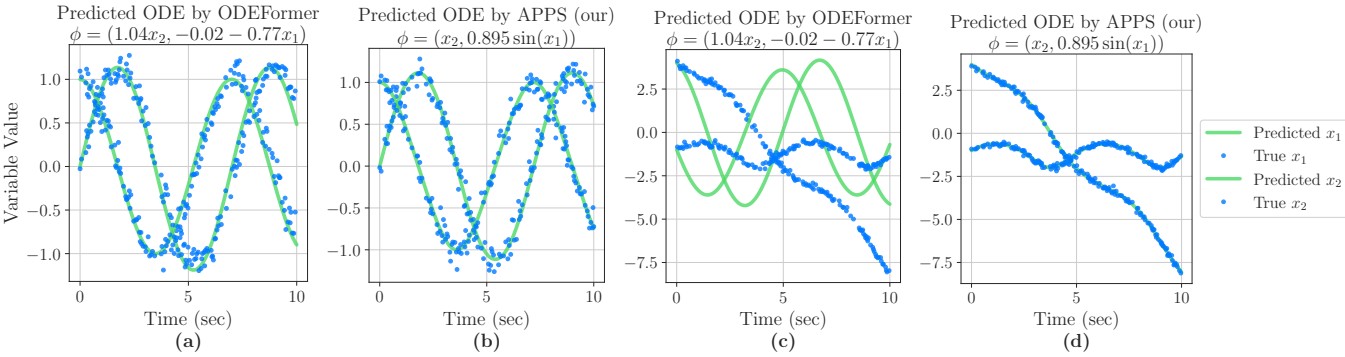

Figure 1: The performance of predicted ODE from passively-learned baseline is heavily influenced by the collected training data while our APPS method is not. The dots represent noisy ground-truth trajectory data, and the lines show predicted values of state variables under identical initial conditions. **(a, b)** Our APPS and the baseline predict accurately for the trajectory starting at $\mathbf{x}_0 = (0, 1)$. **(c, d)** For the trajectory starting at $\mathbf{x}_0 = (4, -1)$, the baseline performs poorly while APPS maintains accuracy.

data query and evaluation module that actively queries the data using sketched phase portraits and evaluates the candidate ODE. In experiments, we evaluate APPS against several popular baselines on two large-scale ODE datasets. 1) APPS achieves the lowest median NMSE (in Table 1 and Table 2) across multiple datasets under noiseless and noisy settings. 2) Compared to other active learning strategies, APPS is more time efficient in benchmark datasets (in Table 3). [1].

## 2  Preliminaries

**Ordinary Differential Equations** (ODEs) describe the evolution of dynamical systems in continuous time. Let vector $\mathbf{x}(t) = (x_1(t), \ldots, x_n(t)) \in \mathbb{R}^n$ be the state variables of the system of time $t$. The temporal evolution of the system is governed by the time derivatives of the state variables, denoted as $\frac{dx_i}{dt}$. The general form of the ODE is written as:

$$\frac{dx_i}{dt} = f_i(\mathbf{x}(t), \mathbf{c}), \quad \text{for } i = 1, \ldots, n,$$

where $f_i$ can be a linear or nonlinear function of the state variables $\mathbf{x}$ and coefficients $\mathbf{c}$. The ODE is noted as a tuple $(f_1, f_2, \ldots, f_n)$ for simplicity in this paper. Function $f_i$ is symbolically expressed using a subset of input variables in $\mathbf{x}$ and coefficients in $\mathbf{c}$, connected by mathematical operators such as addition and cosine functions. For example, we use $(10\sin(x_2), 4\cos(x_1 + 2))$ to represent the ODE $\frac{dx_1}{dt} = 10\sin(x_2), \frac{dx_2}{dt} = 4\cos(x_1 + 2)$.

Given an initial condition $\mathbf{x}_0$, the solution to the ODE is a *trajectory* of state variables $(\mathbf{x}_0, \mathbf{x}(t_1), \ldots, \mathbf{x}(t_k))$ observed at discrete time points $(t_1, \ldots, t_k)$, possibly with noise. The trajectory is noted as $\tau$ for simplicity.

**Phase Portrait** is a qualitative analysis tool for studying the behavior of dynamical systems (Strogatz 2018). Phase portraits are plotted using the state variables $\mathbf{x}$ and their time derivatives $(f_1, \ldots, f_n)$. A curve in the phase portrait is a short trajectory of the system over time from a given initial condition. The arrow on the curve indicates the direction of

change. By examining these curves, we can infer key properties of the system, such as stability, equilibrium points, and periodic behavior. Figure 2(c) shows phase portraits for three different ODEs. These portraits are generated by sampling random initial conditions within the variable intervals and evolving the system for a short time.

**Symbolic Discovery of Ordinary Differential Equations** seeks to uncover the symbolic form of an ODE that best fits a dataset of observed trajectories. According to Gec et al. (2022) and Sun et al. (2023), we are given a dataset of collected trajectories $D = \{\tau_1, \ldots, \tau_N\}$ and a set of mathematical operators $\{+, -, \times, \div, \sin \ldots\}$. Denote $\phi(\mathbf{x}(t), \mathbf{c})$ as a candidate ODE, where $\mathbf{c}$ indicates the coefficients. The objective is to predict the symbolic form of the ODE that minimizes the distance between the predicted and observed trajectories, which is formalized as an optimization problem:

$$\arg\min_{\phi \in \Pi} \frac{1}{|D|} \sum_{\tau \in D} \sum_{i=1}^{k} \ell(\mathbf{x}(t_i), \hat{\mathbf{x}}(t_i)),$$

$$\text{where} \quad \hat{\mathbf{x}}(t_i) = \mathbf{x}_0 + \int_0^{t_i} \phi(\mathbf{x}(t), \mathbf{c}) dt. \tag{1}$$

$\Pi$ is the set of all possible ODEs, trajectory $\tau := (\mathbf{x}_0, \mathbf{x}(t_1), \ldots, \mathbf{x}(t_k))$, $\mathbf{x}(t)$ is the ground-truth observations of the state variable. Trajectory $(\mathbf{x}_0, \hat{\mathbf{x}}(t_1), \ldots, \hat{\mathbf{x}}(t_k))$ is the predicted state variables according to the candidate ODE $\phi$. The predicted trajectory $(\mathbf{x}_0, \hat{\mathbf{x}}(t_1), \ldots, \hat{\mathbf{x}}(t_k))$ is obtained by numerically integrating the ODE from the given initial state $\mathbf{x}_0$ to the final time $t_k$. The loss function $\ell$ computes the summarized distance between the predicted and ground-truth trajectories at each time step. A typical loss is the Normalized Mean Squared Error (NMSE, defined in Appendix D). Except for the above formulation, prior works in symbolic regression use the approximated time derivative as the *label* to discover each expression $f_i$ separately, which is known as gradient matching. We leave the discussion to Related Work.

Recent research explored deep reinforcement learning to discover the governing equations from data (Petersen et al. 2021; Abolafia, Norouzi, and Le 2018; Mundhenk et al. 2021). In these approaches, each expression is represented

---

[1]The code is at https://github.com/jiangnanhugo/APPS-ODE. Please refer to https://arxiv.org/abs/2409.01416 for the appendix.

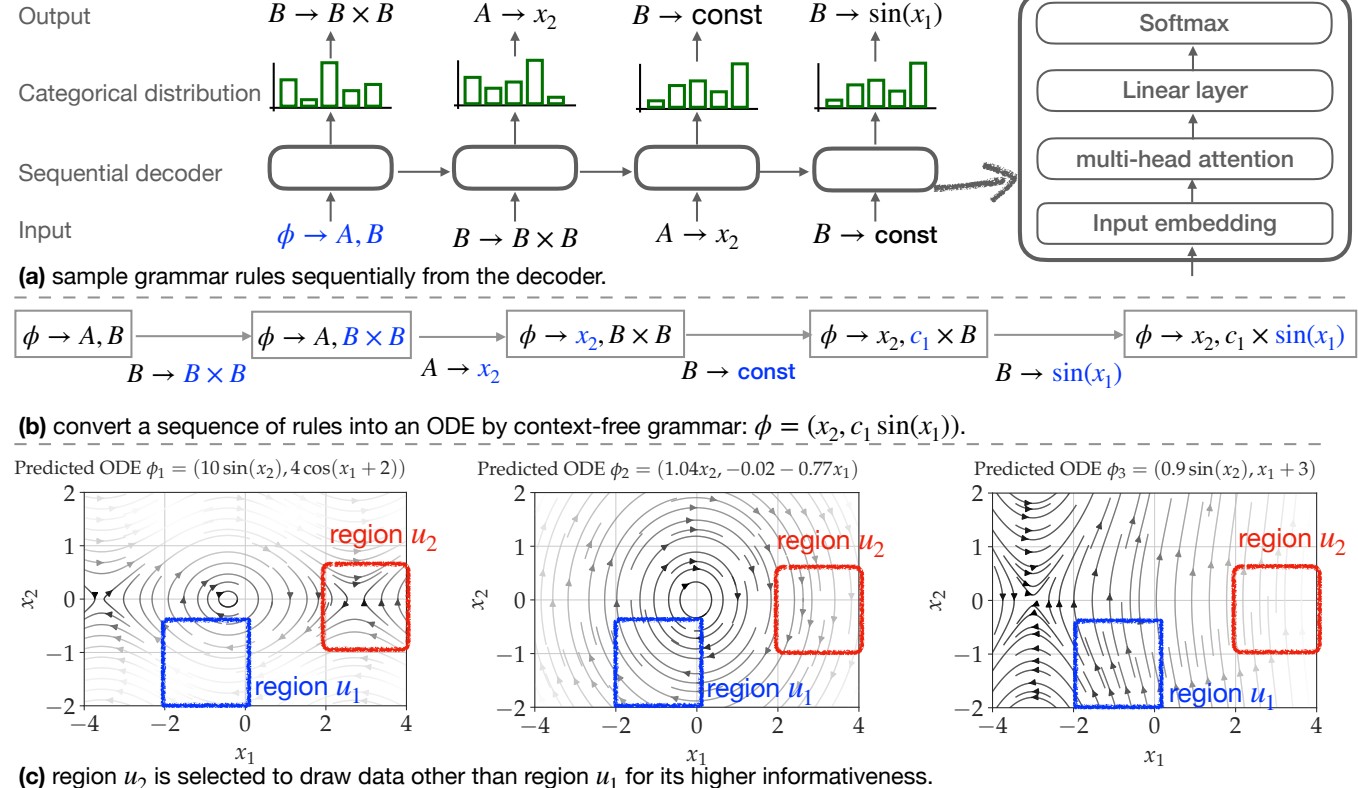

**(a)** sample grammar rules sequentially from the decoder.

**(b)** convert a sequence of rules into an ODE by context-free grammar: $\phi = (x_2, c_1 \sin(x_1))$.

**(c)** region $u_2$ is selected to draw data other than region $u_1$ for its higher informativeness.

Figure 2: The pipeline of APPS for symbolic discovery of ODEs consists of 3 steps: **(a)** ODEs are sampled from the sequential decoder by iteratively sampling grammar rules. The predicted rule at each step serves as input for the decoder in the subsequent step. **(b)** The sampled sequence of grammar rules is converted into a valid ODE with $n = 2$ variables. Each rule expands the first non-terminal symbol, with the expanded parts highlighted in blue colors for clarity. **(c)** The phase portrait for the predicted ODEs (e.g., $\phi_1, \phi_2, \phi_3$) is sketched, and regions with high informativeness, such as $u_2$, are identified to query the new trajectory data. In region $u_2$, $\phi_1$ exhibits a saddle point, $\phi_2$ moves downward, and $\phi_3$ moves upward. In contrast, in region $u_1$, all trajectories move from right to left. Differentiating the predicted expressions is easier in region $u_2$ than in region $u_1$.

as a binary tree, with interior nodes corresponding to mathematical operators and leaf nodes to variables or constants. An ODE with $n$ variables is represented by $n$ trees. The key idea is to frame the search for different ODEs as a sequential decision-making process based on the preorder traversal sequence of expression trees. A high reward is assigned to candidates which fit the data well. The search is guided by a deep sequential decoder, often based on RNN, LSTM, or decoder-only Transformer, that learns the optimal probability distribution for selecting the next node in the expression tree at each step. The parameters of the decoder are trained with the policy gradient algorithm.

## 3 Methodology

**Motivation**

For the task of symbolic discovery of ODEs, we observe that existing methods frequently overfit the training data. This issue is illustrated in Figure 1 using ODEFormer (d'Ascoli et al. 2024), a recent baseline designed to learn ODEs from a fixed training dataset. In the example, the best-predicted ODE is given by $\phi = (1.04x_2, -0.02 - 0.77x_1)$. We evaluate $\phi$ on noisy test trajectories (depicted as blue dots) with two distinct initial conditions. While $\phi$ closely aligns with the trajectory originating at $\mathbf{x}_0 = (0, 1)$, as shown by the green curve, it produces substantial errors for a trajectory starting at $\mathbf{x}_0 = (4, -1)$, where the predicted curve deviates significantly from the ground truth.

This observation motivates us to actively identify informative trajectory data to better differentiate candidate expressions during the learning process. Each trajectory is generated by querying the data oracle with a specified initial condition $\mathbf{x}_0$. An initial condition is deemed *informative* if the resulting trajectory for different candidate ODEs diverges significantly. The key challenge lies in selecting such informative initial conditions from the variable intervals for a given set of candidate ODEs.

For addressing this issue, a common approach in active learning (Golovin, Krause, and Ray 2010) is to maintain a large set of potential initial conditions, evaluate their informativeness, and query the most informative points. However, the butterfly effect in chaos theory (Lorenz 1963) suggests existing works in active learning are not directly applicable. The chaotic behavior states small changes in initial

conditions can lead to drastically different outcomes in dynamical systems. For example, as shown in Figure 2(c), selecting points near $(3, 0)$ (inside the red region $u_2$) for $\phi_1$ can lead to trajectories diverging either towards the top right or the bottom left. Such chaotic behavior necessitates the existing active learning methods to maintain a large set of initial conditions to adequately cover the domain, which becomes infeasible for high-dimensional dynamical systems.

To mitigate this issue, we consider selecting a beam of near-neighbor points rather than individual points. We propose first to select a highly informative region and sample a batch of initial conditions within that region. In this research, the region is represented as an $n$-dimensional cube of fixed width. A region is regarded as *informative* if the majority of sampled initial conditions within it yield informative trajectories for the given candidate ODEs.

Figure 2(c) illustrates our region-based approach using the phase portraits of three candidate ODEs: $\phi_1, \phi_2$, and $\phi_3$. Each curve in the phase portrait represents a short trajectory, with its starting point and direction indicating the initial conditions and the direction of evolution over time. A closer look reveals significant differences in dynamics within region $u_2$ across the ODEs. While the curves in region $u_1 = [-2, 0] \times [-2, 0]$ consistently move from the bottom right to the top left in all phase portraits, the trajectories in region $u_2 = [2, 4] \times [-1, 1]$ exhibit drastically different behaviors. This indicates that trajectories originating from region $u_2$ are more divergent and thus more informative.

**Main Procedure.** The proposed APPS, illustrated in Figure 2, comprises two key components: (1) Deep Sequential Decoder. This module predicts candidate ODEs by sampling sequences of grammar rules defined for symbolic ODE representation. (2) Data Sampling Module. Using the proposed phase portrait sketching, this module selects a batch of informative ground-truth data points.

Throughout the training process, the reward for the predicted ODEs is computed using the queried data, and the decoder parameters are updated via policy gradient estimation. Among all sampled candidates, APPS selects the ODE with the smallest loss value (as defined in Equation 1) as the final prediction.

**Connection to Existing Approaches.** Like d'Ascoli et al. (2024), APPS employs a Transformer-based decoder. However, unlike d'Ascoli et al. (2024), which learns from fixed data, APPS actively queries new data. The learning objective of APPS is inspired by Petersen et al. (2021), where both approaches guide the search for the optimal equation as a decision-making process over a sequence of tokens.

Existing active learning methods, particularly in symbolic regression, have largely overlooked the chaotic behaviors inherent in dynamical systems. For instance, Jin et al. (2023) proposed a separate generative model for sampling informative data, assuming that input data within a small region should exhibit minimal output divergence. However, this assumption fails to hold in the context of dynamical systems. Additionally, Haut, Banzhaf, and Punch (2024) formulated an optimization problem based on the Query-By-Committee (QBC) method in active learning, to find those informative initial conditions. But the optimization needs to maintain a large set of data points, to account for the chaotic behaviors. The rest of the discussion is provided in the Related Work.

## The Learning Pipeline

**Data Assumption.** Our method relies on the assumption that we can query a data oracle $\mathcal{O}$ by specifying the initial conditions $\mathbf{x}_0$ and discrete times $T = (t_1, \ldots, t_k)$. The oracle executes $\mathcal{O}(\mathbf{x}_0, T)$ and returns a (noisy) observation of the trajectory at the specified discrete times $T$. In science, this data query process is achieved by conducting real-world experiments with specified configurations. Recent work (Chen and Xue 2022; Keren, Liberzon, and Lazebnik 2023; Haut, Punch, and Banzhaf 2023) also highlight the importance of having the oracle that can actively query data points, rather than learning from a fixed dataset.

**Expression Representation.** To enable the sequential decoder to predict an ODE by generating a sequence step-by-step, we extend the context-free grammar to represent an ODE as a sequence of grammar rules (Todorovski and Dzeroski 1997; Gec et al. 2022; Sun et al. 2023). The grammar is defined by the tuple $\langle V, \Sigma, R, S \rangle$, where $V$ is a set of non-terminal symbols, $\Sigma$ is a set of terminal symbols, $R$ is a set of production rules and $S \in V$ is the start symbol.

More specifically, each component of the grammar is: 1) For the non-terminal symbols, we use $A$ to represent a sub-expression for $\frac{dx_1}{dt}$ and $B$ to represent a sub-expression for $\frac{dx_2}{7} dt$. For dynamical systems with $n$ variables, we use $n$ distinct non-terminal symbols. 2) The terminal symbols include the input variables and constants $\{x_1, \ldots, x_n, \texttt{const}\}$. 3) The production rules correspond to mathematical operations. For example, the addition operation is represented as $A \rightarrow (A + A)$, where the rule replaces the left-hand symbol with the right-hand side. 4) The start symbol is redefined as "$\phi \rightarrow A, B$", where the comma notation indicates that $A$ and $B$ represent two separate equations in a two-variable dynamical system. Similarly, there will be $n$ non-terminal symbols connected by $n - 1$ comma for $n$-dimensional dynamical system.

Starting from the start symbol, different symbolic ODEs are constructed by applying the grammar rules in various sequences. An ODE is valid if it only consists of terminal symbols; otherwise, it is invalid. Figure 2(b) provides an example of constructing the ODE $\frac{dx_1}{dt} = x_2$, $\frac{dx_2}{dt} = -0.9 \sin(x_1)$ from the start symbol $\phi \rightarrow A, B$ using a sequence of grammar rules. The replaced parts are color highlighted. Initially, the multiplication rule $B \rightarrow B \times B$ is applied, replacing the symbol $B$ in $f_2 = B$ with $B \times B$, resulting in $\phi \rightarrow A, B \times B$. Next, the rule $A \rightarrow x_2$ is applied, yielding $\phi \rightarrow x_2, B \times B$. Iteratively applying the rules, we eventually obtain $\phi \rightarrow x_2, c_1 \times \sin(x_1)$, which corresponds to one candidate ODE $\phi = (x_2, c_1 \sin(x_1))$. The coefficient $c_1 = -0.9$ is obtained when fitting to the trajectory data. The procedure of coefficient fitting is described in Appendix C "Implementation of APPS" section.

**Sampling ODEs from Decoder.** The proposed APPS is built on top of a sequential decoder, which generates different ODEs as a sequential decision-making process. The decoder can be RNN, LSTM, or the decoder-only Transformer. The

input and output vocabulary is the set of allowed rules covering input variables, coefficients, and mathematical operators. Predicting ODEs involves using the decoder to sample a sequence of grammar rules, where each sequence corresponds to a candidate ODE using previously defined grammar. The objective of APPS is to maximize the probability of sampling those ODEs that fit the data well. This is achieved through the REINFORCE objective, where the objective computes the expected reward of ODE to the data. In our formulation, the reward is evaluated on selected data by the phase portrait sketching module.

As shown in Figure 2(a), the decoder receives the start symbol $s_0 = "\phi \to A, B"$ and outputs a categorical distribution $p_\theta(s_1|s_0)$ over rules in the output vocabulary. This distribution represents the probabilities of possible next rules in the partially completed expression. One token is drawn from this distribution, $s_1 \sim p(s_1|s_0)$, which serves as the prediction for the second rule and is used as the input for the next step. At $t$-th step, the predicted output from the previous step $s_t$ is used as the input for the current step. The decoder draws rule $s_{t+1}$ from the probability distribution $s_{t+1} \sim p_\theta(s_{t+1}|s_0, \ldots, s_t)$. This process iterates until maximum steps are reached, with a probability of $p_\theta(s) = \prod_{i=1}^{m-1} p_\theta(s_i|s_1, \ldots, s_{i-1})$. The sampled sequence is converted into an expression following the definition previously described in "Expression Representation".

**Active Query Data with Phase Portrait Sketching.** To evaluate the goodness-of-fit of generated ODEs from the decoder and differentiate which one is better, we propose comparing the phase portrait of predicted ODEs. We sketch the phase portrait using collections of short trajectories, all starting from the same initial conditions and sharing the same time sequence.

Following our discussion in the "Motivation" section, a region is considered informative for distinguishing between two candidate ODEs if their sketched phase portraits differ. To identify such regions, we randomly sample several and sketch the phase portraits for all candidate ODEs within each. The most informative region is then selected, and we query the data oracle (noted as $\mathcal{O}$) for the ground-truth trajectory in that region.

Formally, assume we are given $M$ ODEs, $\{\phi_1, \ldots, \phi_M\}$, and $K$ randomly selected regions, $\{u_1, \ldots, u_K\}$. Each region $u_k$ is a Cartesian product of $n$ intervals, expressed as $u_k = [a_1, b_1] \times \cdots \times [a_n, b_n]$. To sketch the dynamics of candidates in the region $u_k$, we uniformly sample $L$ points in $u_k$, $\mathbf{x}^1, \ldots, \mathbf{x}^L$, as initial conditions. For region $u_k$, the trajectory $\tau_{m,k,l} = (\mathbf{x}^l, \hat{\mathbf{x}}(t_1), \ldots, \hat{\mathbf{x}}(t_k))$ is generated by the expression $\phi_m$, starting from the $l$-th initial condition $\mathbf{x}^l$ and evolving over time according to the numerical integration $\hat{\mathbf{x}}(t_i) = \mathbf{x}^l + \int_0^{t_i} \phi_m(\mathbf{x}(t), \mathbf{c}) \, \mathrm{d}t$ for $t_i \in \{t_1, \ldots, t_k\}$. The resulting $L$ short trajectories form a sketched phase portrait for ODE $\phi_m$ in the region $u_k$.

Two expressions, $\phi_m$ and $\phi_{m'}$, have similar sketches in region $u_k$ if their corresponding trajectories, starting from the same initial condition, are close. Specifically, this occurs when $\sum_{l=1}^{L} \|\tau_{m,k,l} - \tau_{m',k,l}\| \approx 0$. We define the pairwise informative score between $\phi_m$ and $\phi_{m'}$ in region $u_k$ as:

$$\mathtt{IF}(\phi_m, \phi_{m'}, u_k) = \frac{1}{L} \sum_{l=1}^{L} \|\tau_{m,k,l} - \tau_{m',k,l}\|_2^2 \quad (2)$$

The total informative score for a region (denoted as $\mathtt{IF}(u_k)$) is the sum of the pairwise informative scores for every pair of candidate ODEs. The informative score for region $u_k$ is:

$$\mathtt{IF}(u_k) = \frac{1}{M} \sum_{m=1}^{M} \sum_{m'=m+1}^{M} \mathtt{IF}(\phi_m, \phi_{m'}, u_k) \quad (3)$$

We select the region with the highest informative score, denoted $u^* \leftarrow \arg\max_{k=1}^{K} \mathtt{IF}(u_k)$. A batch of $m$ initial conditions, $\{\mathbf{x}_1, \ldots, \mathbf{x}_m\}$, is then sampled from region $u^*$, and the data oracle $\mathcal{O}(\mathbf{x}_i, T)$ is queried with the given initial conditions. The obtained ground-truth trajectories are used to compute the reward function for the objective, which in turn updates the model's parameters. In practice, the relative size of the regions and the number of sampled regions are set as hyper-parameters in the experiments.

**Policy Gradient-based Training.** The REINFORCE objective that maximizes the expected reward is

$$J(\theta) := \mathbb{E}_{s \sim p_\theta(s)}[\mathtt{reward}(s)]$$

where $p_\theta(s)$ is the probability of sampling sequence $s$ and $\theta$ represents the parameters of the decoder. Following the RE-INFORCE policy gradient algorithm (Williams 1992), the gradient *w.r.t.* the objective $\nabla_\theta J(\theta)$ is estimated by the empirical average over the samples from the probability distribution $p_\theta(s)$. We first sample $N$ sequences $(s^1, \ldots, s^N)$, and an unbiased estimation of the gradient of the objective is computed as:

$$\nabla_\theta J(\theta) \approx \frac{1}{N} \sum_{i=1}^{N} \mathtt{reward}(s^i) \nabla_\theta \log p_\theta(s^i)$$

The parameters of the decoder are updated using the estimated policy gradient value. This update process increases the probability of generating high goodness-of-fit ODEs. Detailed derivations are presented in Appendix C.

## 4   Related Work

**AI-driven Scientific Discovery.** Artificial intelligence has increasingly been employed to accelerate discoveries in learning ordinary and partial differential equations directly from data (Brunton, Proctor, and Kutz 2016; Wu and Tegmark 2019; Zhang and Lin 2018; Iten et al. 2020; Cranmer et al. 2020; Raissi, Yazdani, and Karniadakis 2020; Raissi, Perdikaris, and Karniadakis 2019; Liu and Tegmark 2021; Xue et al. 2021; Chen et al. 2018).

**Symbolic Regression for ODEs.** Symbolic regression, traditionally used to identify algebraic equations between input variables and output labels, has been extended to discover ODEs. A key ingredient is gradient matching, which approximates labels for symbolic regression by using finite differences of consecutive states along a trajectory (Sun et al. 2023; Brence, Todorovski, and Dzeroski 2021; Qian,

| | Strogatz dataset ($\sigma^2 = 0, \alpha = 0$) | | | | ODEbase dataset ($\sigma^2 = 0, \alpha = 0$) | | | |
| --- | --- | --- | --- | --- | --- | --- | --- | --- |
| | $n = 1$ | $n = 2$ | $n = 3$ | $n = 4$ | $n = 2$ | $n = 3$ | $n = 4$ | $n = 5$ |
| SPL | 0.787 | 0.892 | 1.921 | 2.865 | 0.867 | 2.17 | 4.75 | 13.16 |
| E2ETransformer | $6.47E{-}4$ | 1.620 | $T.O.$ | $T.O.$ | 0.757 | $T.O.$ | $T.O.$ | $T.O.$ |
| ProGED | 0.129 | 0.666 | 2.68 | 3.856 | 0.317 | 2.134 | $T.O.$ | $T.O.$ |
| SINDy | $1.90E{-}4$ | **0.217** | 1.539 | 4.810 | 0.521 | 2.112 | 8.334 | 52.12 |
| ODEFormer | 0.0303 | 0.9261 | 1.033 | 1.010 | 0.213 | 0.245 | 1.213 | 3.148 |
| APPS (ours) | **2.06E$-$6** | 0.2912 | **1.011** | **0.521** | **0.1318** | **0.1306** | **1.046** | **3.054** |

Table 1: On the *noiseless* datasets with regular time sequence ($\sigma^2 = 0, \alpha = 0$), Median NMSE is reported over the best-predicted expression found by all the algorithms. Our APPS method can discover the governing expressions with smaller NMSE values than baselines, under the noiseless setting. T.O. means termination with a 24-hour limit.

Kacprzyk, and van der Schaar 2022; Gec et al. 2022). Recent methods, such as SINDy and its extensions (Brunton, Proctor, and Kutz 2016; Egan, Li, and Carvalho 2024), leverage sparse regression techniques to directly learn the structure of ODEs and PDEs from data. They perform particularly well with trajectory data sampled at small, regular time intervals, where the approximations closely align with true derivatives.
**Neural Networks Learns Implicit ODEs.** This research direction involves learning ODE implicitly. Early work employed Gaussian Processes to model ODEs (Heinonen et al. 2018). Neural ODEs further advanced the field by parameterizing ODEs with neural networks, enabling training through backpropagation via ODE solvers (Chen et al. 2018). Physics-informed neural networks integrate physical knowledge, such as conservation laws, into the modeling process (Raissi, Perdikaris, and Karniadakis 2019). Meanwhile, Fourier neural operators use neural networks to learn the functional representation (Li et al. 2021).
**Active Learning** aims to query informative unlabeled data to accelerate convergence with fewer samples (Wagenmaker and Jamieson 2020; Mania, Jordan, and Recht 2022; Sener and Savarese 2018; Ash et al. 2020). In symbolic regression, query-by-committee strategies have been explored to actively query data for discovering algebraic equations (Haut, Banzhaf, and Punch 2022; Haut, Punch, and Banzhaf 2023). For example, Jin et al. (2023) proposed a method that learns uncertainty distributions using neural networks and queries data with high uncertainty. However, all these methods largely overlooked the chaotic behaviors inherent in dynamical systems.

## 5  Experiments

This section shows our APPS can find ODEs with the smallest errors (Normalized MSE) among all competing approaches, under noiseless, noisy, and irregular time settings (see Table 1 and Table 2). Compared to the baselines, our APPS data query strategy requires fewer data and attains a better ranking of the TopK candidate ODEs (see Table 3).

### Experimental Settings

**Datasets.** We consider 2 datasets of multivariate variables, including (1) Strogatz dataset (d'Ascoli et al. 2024) of 80 instances, collected from the Strogatz textbook (Strogatz 2018). It is formalized as a benchmark dataset by (d'Ascoli

et al. 2024). (2) ODEBase dataset (Lüders, Sturm, and Radulescu 2022) of 114 instances, containing equations from chemistry and biology. Each dataset is further partitioned by the number of variables contained in the ODE.

We consider 3 different conditions: (1) regular time noiseless condition, (2) regular time noisy condition, and (3) irregular time condition. In the noiseless setting, the obtained data is exactly the evaluation of the ground-truth expression. In the noisy setting, the obtained data is further perturbed by Gaussian noise. We add multiplicative noise by replacing each $\mathbf{x}(t_i)$ with $(1 + \varepsilon)\mathbf{x}(t_i)$, and $\varepsilon$ is sampled from a zero mean multivariate Gaussian distribution with diagonal variances $\text{diag}(\sigma^2, \ldots, \sigma^2)$. The noise rate is determined by $\sigma^2$. For both noiseless and noisy settings, the data points are sampled at regular time intervals. In the irregular time setting, we first generate the regular time sequence and drop a fraction with probability $\alpha$. The rate of time irregularity is determined by $\alpha$.
**Baselines.** We consider a line of recent works for symbolic equation discovery as our baselines. The methods using passive data query strategy are as follows: (1) SINDy (Brunton, Proctor, and Kutz 2016), (2) ODEFormer (d'Ascoli et al. 2024), (3) Symbolic Physics Learner (SPL) (Sun et al. 2023), (4) Probabilistic grammar for equation discovery (ProGED) (Gec et al. 2022), (5) end-to-end Transformer (E2ETransformer) (Kamienny et al. 2022).
**Evaluation.** For evaluating all the methods, we considered 3 different metrics: (1) goodness-of-fit using NMSE, (2) empirical running time of data querying step, and (3) ranking-based distance. The goodness-of-fit using the NMSE indicates how well the learning algorithms perform in discovering symbolic expressions. Given the best-predicted expression by each algorithm, we evaluate the goodness-of-fit on a larger testing set with longer time steps and a larger batch size of data. The median (50%) of the NMSE is reported in the benchmark table. The full quantiles ($25\%, 50\%, 75\%$) of the NMSE are further provided. The remaining details of the experiment settings are in Appendix D.

### Experimental Analysis

**Goodness-of-fit Benchmark.** We summarize our APPS on several challenging multivariate datasets with noiseless data in Table 1. It shows our APPS attains the smallest median NMSE values on all datasets, against a line of current popular baselines. The performance of SPL and E2Etransformer

|  | Noisy Strogatz datasets ($\sigma^2 = 0.01, \alpha = 0$) | | | | Irregular Strogatz dataset ($\sigma^2 = 0, \alpha = 0.1$) | | | |
|---|---|---|---|---|---|---|---|---|
|  | $n = 1$ | $n = 2$ | $n = 3$ | $n = 4$ | $n = 1$ | $n = 2$ | $n = 3$ | $n = 4$ |
| SPL | 0.938 | 1.019 | 2.915 | 3.068 | 0.127 | 0.526 | 3.196 | 4.193 |
| SINDy | $6.4E{-}3$ | 4.152 | 2.498 | 5.21 | $6.66E{-}4$ | 0.472 | 0.827 | 4.163 |
| ProGED | 0.121 | 0.658 | 3.673 | 3.856 | 0.134 | 0.769 | 2.766 | 4.181 |
| ODEFormer | 0.139 | 0.621 | 2.392 | 0.812 | 0.031 | 1.036 | 1.51 | 1.011 |
| APPS (ours) | **7.75E-4** | **0.369** | **1.381** | **0.657** | **1.06E-6** | **0.215** | **1.012** | **0.947** |

Table 2: On the Strogatz dataset, the Median NMSE is reported over the best-predicted expression found by all the algorithms under noisy or irregular time sequence settings.

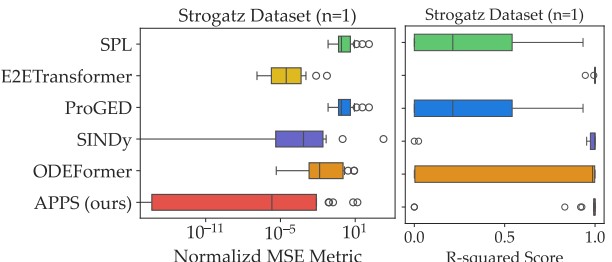

Figure 3: On the selected data (Strogatz dataset with $n = 1$), quartiles of NMSE and $R^2$ scores of the learning algorithms.

|  | Ranking-based distance ($\downarrow$) | Running Time ($\downarrow$) | Peak Memory ($\downarrow$) |
|---|---|---|---|
| APPS (ours) | **0.08** | **5.2** sec | **3.76** MB |
| QbC | 0.13 | 13.4 sec | 51 MB |
| CoreSet | 0.22 | 4.3 sec | 2.74 GB |

Table 3: Ranking comparison with different active learning strategies. APPS shows a smaller ranking-based distance than other strategies, which is better for ranking those best-predicted expressions. Also APPS takes less memory consumption and less computational time because the sketching step itself is lightweight.

drops greatly on irregular time sequences because the approximated time derivative becomes inaccurate when missing the intermediate sequence. Our APPS does not suffer from that because it outputs the predicted trajectory and does not need to approximate the time derivative. Another reason is the decoder with massive parameters can better adapt to actively collected datasets.

**Noisy and Irregular Time Settings.** We examine the performance of predicting trajectories in the presence of noise and irregular time sequences in Table 2. The ground-truth trajectory is subject to Gaussian noise with zero mean and $\sigma^2 = 0.05$, and an irregularly sampled sequence where $50\%$ of evenly spaced points are uniformly dropped. The predicted trajectory by each algorithm is compared against the ground truth, utilizing identical initial conditions. Our APPS still attains a relatively smaller NMSE against baselines under the two settings.

**Quantiles of Evaluation Metrics.** We further report the quantiles of the NMSE metric in Figure 3 to assist the result in Table 1(a). Note that we cut off the negative values as zero when demonstrating $R^2$ score. The two box plots in Figure 3 show the proposed APPS is consistently better than the baselines in terms of the full quantiles $(25\%, 50\%, 75\%)$ of the NMSE metric.

**Benchmark with other Active Strategies.** Two baseline methods using active learning strategy are: (1) query-by-committee (QbC) proposed in (Haut, Banzhaf, and Punch 2022; Haut, Punch, and Banzhaf 2023). (2) Core-Set (Sener and Savarese 2018) proposes to sample diverse data. These methods were originally proposed with different neural networks, thus we evaluate these different active learning methods using the same decoder in our APPS. Current active learning methods are not directly available for evaluation

in our problem setting (in Equation 1), so we re-implement these query strategies with the new problem setting.

The running time of the data querying step measures the efficiency of every active learning algorithm for this task. The ranking-based distance indicates if the ranking of many candidate expressions is exactly the same as evaluated on full data. If the predicted ODEs are ranked in the same order as the full data, then the ranking-based distance (Kendall tau score) will be close to zero.

In Table 3, given a set of 20 predicted ODEs, we compare the TopK ranking (i.e., top 3) of predicted ODEs by each active learning strategy is the same as using full data. We find both our phase portrait and QbC rank those predicted ODEs in proper ranking order. Our APPS takes the least memory to locate the most informative region and is also time efficient because we only pick one region among all the available regions. The QbC takes much more time because it finds every initial condition as an optimization problem over the input variables, which is solved by a separate gradient-based optimizer. CoreSet first runs a clustering algorithm over the ground-truth data and then samples a diverse set of initial conditions from each cluster. So the memory usage of Coreset is mainly determined by the first clustering step.

# 6 Conclusion

In this paper, we introduced APPS, a novel approach for discovering ODEs from trajectory data. By actively reasoning about the most informative regions within the phase portrait of candidate ODEs, APPS overcomes the limitations of passively learned methods that rely on pre-collected datasets. Our approach also reduces the need for extensive data collection while still yielding highly accurate and generalizable

ODE models. The experimental results demonstrate that APPS consistently outperforms baseline methods, achieving the lowest median NMSE across various datasets under both noiseless and noisy conditions.

## Acknowledgments

We thank all the reviewers for their constructive comments. This research was supported by NSF grant CCF-1918327, NSF Career Award IIS-2339844, DOE – Fusion Energy Science grant: DE-SC0024583, and AI Climate: NSF and USDA-NIFA and Cornell University AI for Science Institute.

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
