# OpenReview forum: "Active Symbolic Discovery of Ordinary Differential Equations via Phase Portrait Sketching"
_AAAI.org/2025/Workshop/NeurMAD — AAAI 2025 Workshop NeurMAD Submission_

### Official Review · Reviewer_vy5o · 2024-12-20

**Rating:** 7
**Confidence:** 3

**Review:**

Summary:

The paper introduces APPS, a method for data-driven differential model discovery that employs active learning. APPS avoids the need for the pre computation of large training dataset by actively querying informative data regions, guided by phase portraits of candidate ODEs.

Major concerns:

- The extension of context-free grammar to represent an ODE as a sequence of grammatical rules is not entirely new, but it builds on previous works on symbolic model discovery; it should be better highlighted what the authors' contribution is in this regard.
- Exploration of generalization properties appears limited, particularly with respect to long-term prediction and extrapolation beyond the training time interval. In addition, it would be useful to analyze the generalization capabilities of the framework with respect to discretization steps different from those used during the training procedure.
- A more in depth analysis of training times related to algorithm 1 would be appreciated.
- The test cases considered are not "large-scale." I recommend that the authors apply the proposed framework—or at least discuss and comment on its application—to an ODE derived from the semi-discretization of a PDE, enabling it to handle systems with 1,000 to 10,000 or more DoFs.

---

### Official Review · Reviewer_SZrp · 2024-12-28
**A long full paper with poorly arranged paragraphs**

**Rating:** 5
**Confidence:** 4

**Review:**

Summary

This paper presents a (APPS) method to capture the potential parameters for differential equations from trajectory data. This paper proposes a new active learning method integrated into ODE discovery. This new active learning method incorporates an active path discovery based on more informative data assessment.


Novelty

The method in this paper is very similar to (d’Ascoli, 2024). They have similar pipeline that is made of transformer. (d’Ascoli, 2024) is to read point data as transformer input, by contrary this APPS method is to set mathematical rules as transformer input. The APPS method employs predefined grammar generation rules, by contrary the (d’Ascoli, 2024) is to employ pretrained model.

This paper claims that APPS method is more accurate.

We could say that the (d’Ascoli, 2024) is a global learning, but the APPS method is based on the ranking of local informativeness.

Is this informativeness rating method scalable ? Considering, if there is a large curve, then its local trajectory pattern may be a straight line.


Issue:

1. Some very important explanations and paragraphs are in appendix, particularly the main algorithm is also put into appendix. The main paper is not self-explained without appendix.


2. Although this paper highlighted a new solution to address the “initial condition” dynamic issue, the experiment regarding the improvement on “initial condition” issue are not shown in report.


3. It is necessary to significantly reorganize all the paragraphs for 8-page limit.


Typos:

page 4, figure 2(a), “Categorial distribution” → “Categorical distribution”

page 12, Om = {+, −, ×} → Om = {+, −, ×, /}

---

### Decision · Program_Chairs · 2024-12-30

**Decision:**

Accept

**Comment:**

 This work introduces an interesting method for scientific discovery.